# The Potential Effects of Probiotics and ω-3 Fatty Acids on Chronic Low-Grade Inflammation

**DOI:** 10.3390/nu12082402

**Published:** 2020-08-11

**Authors:** Ashley N. Hutchinson, Lina Tingö, Robert Jan Brummer

**Affiliations:** 1Nutrition-Gut-Brain Interactions Research Centre, School of Medical Sciences, Örebro University, 701 82 Örebro, Sweden; Lina.Tingo@oru.se (L.T.); Robert.Brummer@oru.se (R.J.B.); 2Division of Inflammation and Infection, Department of Biomedical and Clinical Sciences, Linköping University, 581 83 Linköping, Sweden

**Keywords:** probiotics, omega-3 (ω-3) fatty acids, inflammation, gut microbiota, gut-brain axis, dysbiosis

## Abstract

Chronic low-grade inflammation negatively impacts health and is associated with aging and obesity, among other health outcomes. A large number of immune mediators are present in the digestive tract and interact with gut bacteria to impact immune function. The gut microbiota itself is also an important initiator of inflammation, for example by releasing compounds such as lipopolysaccharides (LPS) that may influence cytokine production and immune cell function. Certain nutrients (e.g., probiotics, ω-3 fatty acids [FA]) may increase gut microbiota diversity and reduce inflammation. *Lactobacilli* and *Bifidobacteria*, among others, prevent gut hyperpermeability and lower LPS-dependent chronic low-grade inflammation. Furthermore, ω-3 FA generate positive effects on inflammation-related conditions (e.g., hypertriglyceridemia, diabetes) by interacting with immune, metabolic, and inflammatory pathways. Ω-3 FA also increase LPS-suppressing bacteria (i.e., *Bifidobacteria*) and decrease LPS-producing bacteria (i.e., *Enterobacteria*). Additionally, ω-3 FA appear to promote short-chain FA production. Therefore, combining probiotics with ω-3 FA presents a promising strategy to promote beneficial immune regulation via the gut microbiota, with potential beneficial effects on conditions of inflammatory origin, as commonly experienced by aged and obese individuals, as well as improvements in gut-brain-axis communication.

## 1. Introduction

Acute inflammation is a normal part of the immune response; an essential coordination of the chemical messengers, antibodies, and immune cells at sites of injury or infection [1]. However, states of chronic low-grade inflammation are associated with inflammatory processes that last for an extended period of time, with the risk of triggering a variety of diseases in the organ systems beyond those impacted by the immediate threat (e.g., neurodegeneration, type 2 diabetes) [2]. Chronic low-grade inflammation can be caused by recurrent episodes of acute inflammation, autoimmune disorders, environmental exposure, and defects in immune cell functioning [3].

In addition, chronic low-grade inflammation has been linked to an imbalance in the gut microbiota [2].The digestive system harbors the largest population of microorganisms in the body, where intestinal bacteria interact with one another and with epithelial and mucosal immune cells to maintain immunological homeostasis [4]. The microbiota is part of the body’s communication system integrating the gastrointestinal, immune, and nervous systems, and its stability and adequate function are necessary for maintaining healthy levels of inflammation [5]. Commensal symbionts compete with possible pathobionts for available resources, and these interactions seem to help maintain a healthy equilibrium of the intestinal microbial population [6,7,8]. The digestive tract also contains a large number of immune cells, and a “healthy” microbiota (i.e., a microbiota in a beneficial state of balance [also called eubiosis]), can positively impact immune system functioning [4,9]. Alternatively, microbiota dysbiosis has been linked to inflammatory processes partly through the release of lipopolysaccharides (LPS), which influence cytokine production and activate immune cells [10]. LPS are found in the outer membrane of Gram-negative bacteria and are linked to metabolic endotoxemia and a higher risk of obesity and diabetes [11]. The release of LPS and increase in immune activation may also impair epithelial barrier function [12] and lead to the development of a “leaky gut,” which triggers pro-inflammatory reactions and worsens the intestinal and systemic inflammatory state [13,14,15,16]. The role of the gut microbiota and intestinal permeability in inflammation suggests an alteration in the crosstalk between gut immune cells and commensal bacteria underlying chronic inflammatory conditions [2], representing a microbiota-inflammation duology [10,17].

The microbiota-inflammatory duology might have particular importance in certain patient groups such as aged and obese individuals. The overall increase in circulating levels of pro-inflammatory molecules observed in aging is a commonly known phenomenon described as inflammaging (Figure 1) [2].

Inflammaging is characterized by a vicious cycle of microbiota dysbiosis, intestinal permeability, LPS release, and subsequent activation of the immune system and increases in reactive oxygen species and cellular damage, which can lead to the development of age-related chronic diseases [2]. Altering the gut microbiota with health-promoting bacteria in older individuals has been shown to positively affect immune functions that are important for maintaining an optimal immune response, which declines with aging, such as slowing the aging of T lymphocytes and increasing immune cells that respond to acute antigen exposure [18]. Additionally, over-nutrition can lead to excess fat depositions and cause an infiltration of macrophages and immune cells into adipose tissue, setting off a vicious cycle of unrestricted secretion of cytokines that produce a similar state of chronic low-grade inflammation in obese individuals [19]. The pro-inflammatory state in obese individuals is suggested to be driven by gut microbiota dysbiosis [10]. Differences between the microbiota of obese and lean individuals may lead to variations in the ability to utilize and store energy in adipose tissue [20]. Short-chain fatty acids (SCFAs) produced by gut bacteria, for example, activate mechanisms that regulate energy uptake by white adipose tissue and energy expenditure by the liver, muscles, and other tissues, suggesting an important role in post-prandial nutrient utilization and energy expenditure [21]. Furthermore, dysregulation of gastrointestinal microbiota and altered immune cell function suggest an enteric origin of metabolic disorders such as the type 2 diabetes phenotype [22]. Also, functional gastrointestinal disorders such as irritable bowel syndrome (IBS) and functional dyspepsia are in part thought to be mediated by similar interactions [23].

Moreover, certain nutritional factors are important to the immune system’s functioning and could potentially impact inflammatory processes in a way that leads to either beneficial immune regulation and maintained health or, conversely, to the development of chronic disease [5]. In particular, probiotic bacteria, prebiotic fiber [10], and omega-3 (ω-3) fatty acids [24] have been suggested to serve as positive modulators of this nutrition-inflammation coalition. These dietary components interact with bacterial organisms in the gut, modulating the release of metabolites that signal to a variety of bodily systems (e.g., the immune system). This narrative review describes the impact of probiotics and ω-3 fatty acids (separately and combined) on chronic low-grade inflammation, with a particular focus on their potential utilization to improve health in patient groups that commonly suffer from increased systemic inflammation, such as aged and obese individuals.

## 2. Modulating the Gastrointestinal Microbiota: The Effect of Environmental Exposure and Dietary Supplementation

The intestinal microbiota population is affected by a number of external factors, including early bacterial exposure [10]. This early acquisition of a diverse range of gut bacteria has a role in the immune system’s maturation [25]. In childhood, the composition of the gut microbiota is mainly determined by maternal microbial transfer during birth, which varies based on the method of delivery (e.g., vaginal delivery or cesarean section [CS]) and post-natal nutritional exposure (i.e., to exclusive breastfeeding or formula and timing/composition of solid food introduction) [10,26,27,28,29,30]. Furthermore, microbial exposure in utero imprints the fetal microbiota and immune system in preparation for the microbial exposure during vaginal delivery and lactation [28,31]. In newborns, the lack of exposure to a diverse range of healthy bacteria can negatively impact the immune system’s functioning and potentially lead to the development of certain conditions, such as allergic disease [28,32]. An examination of fecal samples of a group of 7-year-old children revealed an abnormal development of the intestinal microbiota in those born via CS, in particular, lower levels of *Clostridia* [33]. Additionally, infants born via CS may have delayed colonization of *Bacteroides* and *Bifidobacterium* species and lower bacterial diversity during the first 2 years of life [27].

In adulthood, the composition of the intestinal microbiota is strongly dependent on factors such as long-term dietary patterns, antibiotic use, exercise, and psychological stress [5,10,24,25,34]. Because alterations in the intestinal microbiota have been linked to health and disease, many studies have examined ways to alter the microbiota composition through diet and supplementation [9]. Diet is a key factor in determining the activity and composition of the gut microbiota, and dietary changes have been estimated to explain as much as half of the structural variations in microbiota composition [10]. Introducing beneficial bacteria by consuming probiotic supplements has been shown to be especially promising [24]. For example, the probiotic bacteria *Lactobacilli* and *Bifidobacteria*, among other strains, have been widely studied and have been shown to improve intestinal barrier function and prevent hyperpermeability via a variety of mechanisms including Toll-like recptor-2 mediated-immune-modulating and anti-inflammatory effects, by promoting the production of compounds that support intestinal barrier function, such as butyrate, and by eliciting positive structural changes in Zo-1 and occludin as part of the tight junction protein complexes [35]. Thus, they may also decrease transfer of LPS [5,36]. In addition, previous research suggests that probiotic bacteria may have a role in forming certain neuroactive compounds. These compounds act similarly to neurochemicals that have roles in reducing inflammation [37] and facilitating gut-brain axis communication. For instance, γ-aminobutyric acid (GABA) has been isolated from *Bifidobacteria*, and GABA and acetylcholine have been isolated from *Lactobacillus* strains [5,37,38]. The evidence suggests that the composition of the microbiota is driven by a number of factors and has far-reaching health implications, including effects on inflammatory processes and the central nervous system (CNS).

## 3. Human Health and the Gut–Brain Axis

The gut microbiota metabolizes nutrients consumed through the diet into numerous peripherally- and centrally-acting compounds [10]. These compounds include SCFAs and neurotransmitters (e.g., GABA and serotonin [5-HT]), which interact with the immune system and also affect brain function. The communication pathways by which SCFAs and neurotransmitters impact the brain include the vagus nerve, systemic circulation, and secondary hormone formation [10,39]. The direct and indirect connections between the enteric nervous system and the CNS, known as the gut-brain axis, can impact psychological and cognitive functioning (Figure 2) [13,14].

Gut–brain axis dysregulation seems to be exacerbated by alterations in the composition and functioning of the gut microbiota and the level of inflammatory mediators, which might increase the risk for metabolic and psychiatric disorders [15]. Modifying the composition of the gut microbiota thus has several potential positive effects on health by improving communication through the microbiota–gut–brain axis [24,39]. Psychological stress seems to be a potent modifier of inflammation in the gut by increasing intestinal epithelial permeability, adversely affecting immune regulation and influencing the enteric microflora [17]. In vivo testing has shown that early life exposure to stress decreases gut microbiota diversity and may induce long-term compositional changes [10,13]. In adulthood, chronic stress adversely affects gut microbiota homeostasis and composition, decreasing *Bacteroides* and *Clostridium* subspecies with subsequent increases in inflammation. This further compromises gut health and physiology by altering intestinal barrier permeability, leading to local immune activation and increased LPS circulation [8,10]. Moreover, functional gastrointestinal disorders are closely linked to stress, and one plausible mechanism for this connection is a brain–mast cell interaction and the association with inflammatory processes [40]. Mast cells in the digestive tract serve as end-effectors of gut–brain axis activity and are an important component in allergic responses to exogenous antigens by acting in concert with the pro-inflammatory immune cell immunoglobulin E to increase the release of mast cell mediators [41].

The gut microbiota may further be linked to neurological dysfunction through the secretion of pro-inflammatory cytokines, which are likely to play an important role in neuroinflammation [42] and may also be linked to the development of neurological autoimmune diseases, such as multiple sclerosis [43]. Dysregulation of the gut-brain axis, as well as chronic intestinal inflammation and increased permeability, lead to additional pro-inflammatory factors (e.g., interleukin 8) entering the CNS [25,44,45]. Moreover, low-quality diets, which are high in saturated fatty acids, red meat proteins, sugar, and salt and low in fiber, fruits, and vegetables, induce an inflammatory response partly mediated by the gut microbiota that can alter neurological functioning [5]. Several studies have observed a relationship between dysbiosis and increased susceptibility to psychiatric and neurologic pathologies [5]. These changes can lead to the development of neurodegenerative conditions, including Alzheimer’s disease, Parkinson’s disease, and stroke, and psychiatric diseases such as anxiety disorders, depression, and autism [5]. Additionally, dysbiosis has been observed in inflammatory gastrointestinal conditions, such as IBS, that is related to systemic and intestinal inflammatory tone and may further contribute to the development of depression [15]. Bacterial translocation due to a leaky gut has also specifically been shown to contribute to the pathophysiology of major depressive disorder [46]. Furthermore, inflammation and microbiota dysbiosis may alter tryptophan metabolism by promoting indoleamine-2,3 mediated conversion through the kynurenine pathway, leading to a reduction in the formation of 5-HT [47]. Tryptophan and 5-HT are heavily involved in the gut-brain-microbiota axis, [48] and gut inflammation is marked by the altered levels of neurotransmitters, including 5-HT, that are observed in depressed patients [49]. Tryptophan is also associated with immune homeostasis, and once absorbed in the gut, it can cross the blood-brain barrier, where it is a prerequisite for 5-HT synthesis [49].

The molecular mechanisms underlying the crosstalk between the enteric system and CNS in depression are not fully understood; however, inflammation, immune cells and gut bacteria seem able to alter mood and affect neurochemicals in the brain [15,16,49]. Interventions targeting the intestinal microbiota have shown some promising results in affecting the gut–brain axis. There is a robust body of evidence suggesting that probiotic supplements have positive effects on major depression that are driven by increased availability of serotonin and reductions in inflammatory biomarkers [50]. In particular, a randomized, controlled trial (RCT) from Tillisch et al. in a small group of healthy women who were administered *Bifidobacterium animalis* subsp. *Lactis, Streptococcus thermophiles, Lactobacillus bulgaricus, and Lactococcus lactis* subsp. *Lactis* twice daily for 4 weeks demonstrated positive changes in brain activity and emotional regulation [51]. Another RCT conducted in adults with major depressive disorder reported that 8 weeks of administering *Lactobacillus acidophilus* and *Lactobacillus casei* combined with *Bifidobacterium bifidum* significantly reduced depressive symptoms in relation to placebo [52]. Additionally, this combination of probiotic strains significantly reduced metabolic, inflammatory, and oxidative stress biomarkers. These data suggest that there are multiple strategies that may be used to promote human health via modulation of the gastrointestinal microbiota and by reducing chronic low-grade inflammation, including the use of certain dietary supplements, as will be further discussed below.

## 4. Microbiota and Immune Modulation by Probiotics, Prebiotics, and Ω-3 Fatty Acids

Probiotic supplements directly impact the composition of the gut microbiota by introducing healthy bacteria, while dietary fiber and supplements containing indigestible structural carbohydrates can promote the development of a diverse gut microbiota by supporting the growth of certain bacterial species [10,53]. Fibers with prebiotic properties, such as inulin, *β*-glucans, and oligofructose, are metabolized by and modulate the activity of specific bacterial strains in the gut microbiota [5,53]. Therefore, fiber may offer a therapeutic benefit by altering the composition and diversity of the gut microbiota, as well as modifying inflammatory markers [9]. Diets low in microbiota-accessible prebiotic fiber reduce the diversity of the microbiota, while high-fiber, low-sugar, and low-fat dietary patterns are linked to increases in microbiota diversity [10]. Importantly, the metabolic rate of prebiotic compounds determines SCFA levels in the proximal versus distal colon, which can affect their beneficial effects in the different segments of the gastrointestinal tract [54]. SCFA receptors are expressed on immune cells, indicating possible links of SCFAs to improvements in immunity, e.g., via regulation of T-cell function [39]. In addition, increased concentrations of fermentable fiber in the diet can, via increased production of SCFAs subsequently reaching the circulation, induce far-reaching beneficial metabolic effects in organs distant from the intestine, such as adipose tissue, brain, and liver [5,39].

In addition to microbial and immune modulation by pro- and prebiotics, the long-chain ω-3 fatty acids eicosapentaenoic acid (EPA) and docosahexaenoic acid (DHA) show similar positive effects. EPA and DHA, for example, have anti-inflammatory properties that may help curtail metabolic syndrome [55,56] and have been shown to consistently lower C-reactive protein, tumor necrosis factor (TNF), and interleukins in healthy, non-obese individuals [55]. Dietary supplementation with EPA and DHA also leads to their incorporation into brain phospholipids, in particular in the frontal, parietal, and occipital lobes [19,57]. Furthermore, higher plasma levels of ω-3 fatty acids have been associated with improved cognitive functioning in older adults [58]. Additionally, ω-3 fatty acids and probiotics can significantly reduce inflammatory biomarkers in middle-aged and older adults [1,59] and may also influence the gut-brain axis by interacting with and influencing the gut microbiota [24]. There is a growing body of evidence in the literature describing the potential of modulating chronic low-grade inflammation through the use of probiotic and ω-3 fatty acid supplements and the potential effects these nutritional interventions might have on the gut–brain axis and general health status of humans. The results of some key studies in this area are further described in the following sections.

### 4.1. Lactobacillus, Bifidobacterium, and Chronic Low-Grade Inflammation

As discussed in the sections above, probiotics exert beneficial effects on intestinal tissue by modifying the gut microbiota population, adhering to the mucosa, strengthening the epithelial barrier, and regulating the immune system and inflammation [60]. *Lactobacilli* and *Bifidobacteria* are among the phyla and genera of eubacteria that constitute the commensal intestinal microbiota [5]. These strains have been widely studied and seem to have several beneficial effects on human health, such as alleviating alterations in gut permeability and LPS-induced chronic low-grade inflammation throughout the body and in the CNS [5]. Some of the key research that has been conducted using these strains in areas that are related to reductions in chronic low-grade inflammation, including through the gut–brain axis, are summarized in the following sections.

### 4.2. Lactobacillus

Lactic acid bacteria cover a varied range of genera, including *Lactobacillus, Enterococcus, Lactococcus, Pediococcus, Streptococcus, Tetragenococcus, Vagococcus, Leuconostoc, Oenococcus, Carnobacteria,* and *Weissella*, constituting a diverse group often misleadingly labeled as *Lactobacilli* [61]. The central role of *Lactobacilli* within the microbiota is to provide local release of antimicrobial compounds, raising the question of whether every strain of these genera is advantageous.

*L. acidophilus* NCFM and *paracasei* Lpc-37 survive passage through the gastrointestinal tract, adhere to intestinal cells, inhibit mucus adherence of a variety of pathogens, and have the potential to inhibit growth of prokaryotic and eukaryotic pathogens [37,62]. Also, various *Lactobacilli* strains have been shown to adhere to intestinal cells and inhibit and displace mucosal pathogens [63]. Regular use of *L. delbrueckii* subsp. *bulgaricus* 8481, for example, has been shown to prevent cytomegalovirus reactivation in the elderly, indicating that consumption of this probiotic strain could counteract some hallmarks of immunosenescence related to T-cell immunity [2,18]. Furthermore, *Lactobacillus rhamnosus* GG has been shown to reduce pro-inflammatory TNF-α production in peripheral blood mononuclear cells in healthy subjects [64,65]. In addition, a combined dose of 3 *Lactobacillus* strains (*L. paracasei* DSM 13434, *L. plantarum* DSM 15312, and *L. plantarum* DSM 15313) has been shown to reduce pro-inflammatory cytokines in the digestive tract of subjects with neuroinflammation [44,66], which may be a highly relevant effect to be considered in aging populations. *L. plantarum* improved cognitive performance, in particular, attention, and positively impacted levels of brain-derived neurotrophic factor among older individuals with mild cognitive impairment [67].

### 4.3. Bifidobacterium

*B. lactis* Bl-04 has been reported to enhance the immune responsivity of mucosal surfaces and has shown promising benefits for healthy adults in reducing the risk for upper respiratory tract infections [68]. *B. longum* subsp. *infantis* also decreases LPS concentrations, which may reduce inflammatory cytokine production [69], and *B. infantis* 35624 may reduce cytokine- and T-cell–mediated inflammation [70]. These effects are likely to improve intestinal barrier and immune function and may also have a role in alleviating functional gastrointestinal disorders such as IBS, which also has a relationship to the gut–brain axis; IBS patients have been shown to have gut microbiota dysbiosis, as exhibited by 1.5-fold decreases in the number of *Bifidobacteria* compared with healthy controls [71]. IBS symptoms are also connected with an abundance of groups of *Firmicutes*, which correlates with decreases in *Bifidobacteria* [71]. Hence, addition of potent *Bifidobacteria* supplements could potentially benefit this patient group in particular.

### 4.4. Lactobacillus and Bifidobacterium in Combination

Combining probiotic bacterial strains may produce synergistic effects, conferring additive benefits, with the exception of rare instances of antagonistic reactions between strains [62]. *L. rhamnosus* GG, *B. animalis* subsp. lactis, and *Propionibacterium freudenreichii* subsp. *shermanii* all have anti-inflammatory properties, as indicated by observed reductions in levels of C-reactive protein and pro-inflammatory cytokines in healthy adults [64]. Furthermore, consumption of a combination of *L. paracasei* Lpc-37 and *acidophilus* 74-2 and *B. animalis* subsp. *lactis* 420 reduced the symptoms of adult atopic dermatitis by affecting peripheral immune signaling and modulating the gut microbiota, which contributed to improvements in health and immune signaling pathways [72]. A combination of *L. helveticus* R0052 and *B. longum* R0175 has also been shown to reduce psychological stress through the gut-brain axis [73].

*Bifidobacterium lactis* Bi-07 combined with *L. acidophilus* NCFM have been shown to improve digestive health by binding with human-derived proteins (e.g., plasmin), and interacting with the host [62]. This combination has also been found to modestly reduce bloating in individuals with functional bowel disorders [74] and to prevent upper respiratory tract infections in adults [68]. Additionally, *B. lactis* Bi-07 combined with *L. acidophilus* NCFM produced beneficial effects on immune functioning in children, including reducing the risk for illness, antibiotic prescriptions, and missed school days [75]. Furthermore, a vegetable capsule formulation containing various *Bifidobacterium* and *Lactobacillus* strains together with *S. thermophilus* DSM 24731 may reduce low-grade inflammation in older adults, as well as lower homocysteine concentrations and increase folate and vitamin B_12_ levels, which could potentially reduce the risk of a range of age-related conditions, including neurological conditions, through the gut–brain axis [76].

## 5. Fatty Acids and Chronic Low-Grade Inflammation

In addition to the effects of probiotics on inflammation, the mechanisms by which ω-3 fatty acids reduce inflammation have been studied extensively. In terms of the effects of ω-3 fatty acids in metabolic disorders, free fatty acid receptor-4 (FFAR4) signaling has been identified as the main route by which ω-3 fatty acids mediate anti-inflammatory and insulin-sensitizing effects to curb metabolic syndrome [56]. Evidence, however, exists for alternative routes independent of FFAR4 for the beneficial effects of ω-3 fatty acids in immune regulation. White adipose tissue, for example, seems to have an important role in metabolizing a series of lipokines from ω-3 fatty acids that are involved in the beneficial anti-inflammatory effects attributed to DHA in both mice and humans [77].

Ω-3 fatty acids positively influence adipose tissue biology and metabolism by decreasing the storage of ingested fat, normalizing secretion of cytokines, and achieving adipose-specific blunting of inflammation [19]. Twenty-four weeks of treatment with EPA 320 mg and DHA 200 mg daily showed beneficial effects on waist circumference, glucose metabolism, glycosylated hemoglobin, leptin, leptin/adiponectin ratio, and lipid profile among individuals with type 2 diabetes [78]. Long-chain ω-3 fatty acids have further been shown to positively affect eicosanoid production and gene expression (e.g., peroxisome proliferator-activated receptor [PPAR] y and nuclear factor kappa B [NF-kB]), thereby lowering pro-inflammatory cytokine production by different cell types [79]. During inflammatory states, cytokines stimulate endothelial cells to develop adhesion receptors and effector cells are recruited by pro-inflammatory cytokines and chemokines [80]. Ω-3 fatty acids can further decrease adhesion of cytokines on endothelial cell receptors [80,81] and may hence positively attenuate the inflammatory response. Intake of EPA and DHA may also increase proportions of these fatty acids in the inflammatory cell phospholipids, replacing arachidonic acid and thereby reducing synthesis of inflammatory eicosanoids [2]. Supplementation with ω-3 fatty acids also allows EPA and DHA to be incorporated into adipose tissue and brain phospholipids, which can reduce obesity-related inflammation [19,82].

Dose-dependent actions of ω-3 fatty acids on immune response have not been well described, potentially because doses as high as 2 g per day may be required to achieve an effect [81]. The beneficial effects of ω-3 fatty acids may be more pronounced in overweight subjects with underlying inflammation, and those effects are likely attributed to metabolic factors related to obesity rather than to the obesity itself [83]. If compared with EPA, DHA has been shown to be a more potent modulator of inflammatory markers in otherwise healthy subjects with abdominal obesity and low-grade systemic inflammation [84]. Moreover, ω-3 fatty acid supplementation potentially has a positive effect on inflammation-related conditions such as Alzheimer’s disease, hypertriglyceridemia, and diabetes [85], all of which are highly relevant in the aging population.

## 6. Synergism between ω-3 Fatty Acids and Probiotics

There is a growing body of evidence to suggest that the combination of probiotic strains with ω-3 fatty acids may provide an additive health benefit to when either supplement is administered alone [86]. Ω-3 fatty acids can act as prebiotics in the gut and have been linked to improvements in composition and diversity of gut microbiome in middle-aged and elderly women [87]. Although the mechanism is not well understood, ω-3 fatty acids also seem to interact with the gastrointestinal microbiota [24] and may, for example, increase levels of LPS-suppressing bacteria (i.e., *Bifidobacteria)* and decrease LPS-producing bacteria (i.e., *Enterobacteria*) [24]. Dysbiosis of the microbiota *Firmicutes/Bacteroidetes* ratio is associated with increases in gut permeability, depression, insulin resistance, and obesity; however, ω-3 fatty acids can aid in restoring the *Firmicutes/Bacteroidetes* ratio, increasing *Lachnospiraceae* taxa and thereby promoting the production of the SCFA butyrate [24]. EPA and DHA have further been described to influence the gut microbiota composition favorably by supporting a lean phenotype [11].

Fatty acids and the gut microbiota have shared routes in the inhibition and activation of the immune system. Saturated fatty acids (SFAs) are related to negative changes in the gut microbiota, leaky gut, weight gain, and pro-inflammatory status, while ω-3 fatty acids have anti-inflammatory properties, can prevent a leaky gut, and positively modulate host microbial ecosystems [88]. The anti-inflammatory effects exerted by ω-3 fatty acids will possibly benefit microbiome composition due to products of DHA metabolism [87]. While SFAs enhance intestinal permeability, supplementation with DHA can help maintain gut epithelial cell barrier integrity despite excess dietary SFAs [89,90]. LPS increase as a result of a high-fat diet, possibly mediated by gut permeability, which enhances LPS absorption and, thereby, raises the risk for metabolic endotoxemia, adipose tissue inflammation, and metabolic disorders [90]. Metabolic endotoxemia from gut dysbiosis is central to the pathogenesis of chronic low-grade inflammation [91].

Supplementation with both ω-3 fatty acids and probiotics has further been associated with positive metabolic outcomes (e.g., glucose levels, insulin metabolism) during pregnancy, which might be associated with the pregnancy phenotype, i.e., excess adiposity, insulin resistance and increased inflammatory status [92]. This also supports that similar benefits may be evident in non-pregnant populations, such as in diabetic and pre-diabetic subjects, since pregnancy somewhat resembles type 2 diabetes from a metabolic perspective [92].

Furthermore, the connection between inflammation and metabolic disease is an emerging area of research and presents an opportunity to explore the use of ω-3 fatty acids combined with probiotic bacteria for their immunomodulating effects [92]. An RCT of 60 overweight adults showed a combination of ω-3 fatty acids and VSL-3 probiotic formulation (comprising *Bifidobacteria, Lactobacilli,* and *S. thermophilus*) resulted in greater improvement in insulin sensitivity than probiotic administration alone [86].

Moreover, certain dietary habits and nutritional supplements (e.g., probiotics, ω-3 fatty acids) can increase the diversity of gut microbiota composition and thereby reduce low-grade inflammation [9,10,24]. In an older population (≥65 years), a diverse diet including more fruits, vegetables, whole grains, and dairy products, for example, seems to promote a more diverse microbial population and also relates to better overall health. Therefore, dietary supplements designed to promote a healthy gut microbiota may be particularly useful in maintaining health in this age group [93]. A small controlled trial that evaluated the effects of ω-3 fatty acids from flax seed and wheat germ oil combined with *Lactobacillus* + *Lactococcus*, *Bifidiobacterium*, and *Acetobacter* in adults with non-alcoholic fatty liver disease reported reductions from baseline in liver fat, improvements in serum lipid levels and metabolic profile, and reduced systemic inflammation [94]. Moreover, colorectal cancer patients undergoing chemotherapy receiving a supplement containing different *Lactobacilli* and *Bifidobacteria* strains combined with ω-3 PUFAs experienced improved quality of life and inflammatory biomarkers [95]. Hence, as discussed here, combining probiotics and ω-3 fatty acid supplements may be a particularly beneficial strategy, as they seem to promote health in various areas through synergistic effects.

## 7. Conclusions

The number of scientific studies suggesting a positive impact of ω-3 fatty acids and probiotics (individually and combined) on low-grade inflammation is growing, and increasing evidence points towards health-promoting effects from these supplements throughout the life span (Table 1). These effects extend beyond gastrointestinal health to include positive effects on the gut-brain axis and neurological functioning. There are early indications that ω-3 fatty acids can act as prebiotic compounds and help to establish a healthy gut microbiota population. Strategies to modulate and manipulate the gut microbiota through supplements such as ω-3 fatty acids, prebiotics, and probiotics may enhance immune regulation and prevent and/or treat chronic disease. Hypothetically, combining ω-3 fatty acids with probiotics offers a promising strategy to prevent the development of low-grade inflammation as well as offering non-pharmaceutical treatment modalities, which might be especially relevant in patient groups that suffer from increased systemic inflammation, such as aged and obese individuals. However, this research field is still in need of well-conducted and properly controlled clinical trials to further support this hypothesis.

## Figures and Tables

**Figure 1 nutrients-12-02402-f001:**
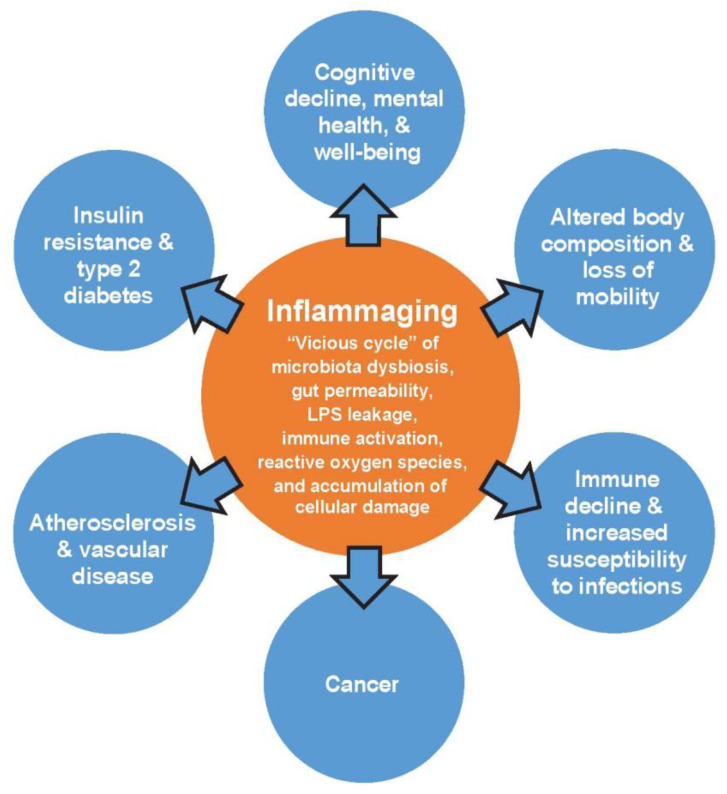
Central role of inflammaging in chronic conditions of aging [2]. LPS, lipopolysaccharides. Adapted from Calder P.C. et al. *Ageing Res Rev* 2017, *40*, 95–119, licensed under CC BY-NC-ND 4.0.

**Figure 2 nutrients-12-02402-f002:**
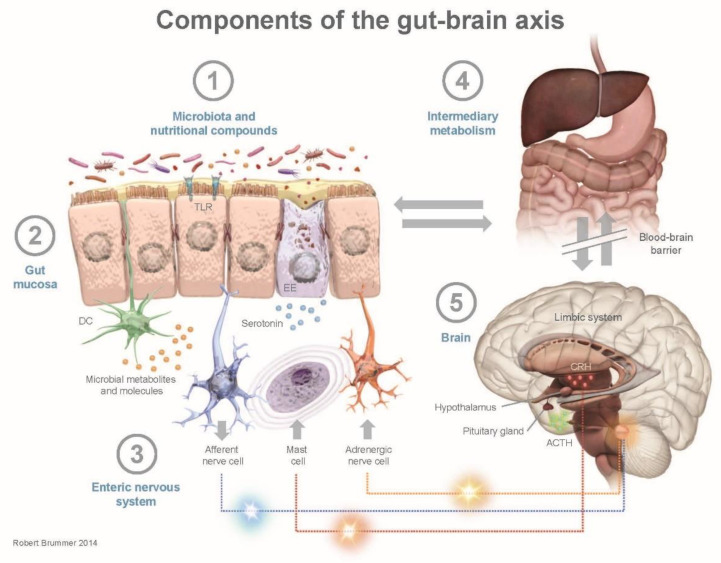
Components of gut-brain axis communication. ACTH, adrenocorticotropic hormone; CRH, corticotropin-releasing hormone; EE, enterorendocrine cell; DC, dendritic cell; TLR, toll-like receptor.

**Table 1 nutrients-12-02402-t001:** Beneficial effects of probiotics, ω-3 fatty acids, and probiotics + ω-3 fatty acids.

Probiotics	ω-3 Fatty Acids	Probiotics + ω-3 Fatty Acids
Positive effects on epithelial barrier function	Normalize secretion of cytokines	Associated with positive metabolic outcomes during pregnancy
May play a role in forming neuroactive compounds	Adipose-specific blunting of inflammation and decreased storage of ingested fat	Improved insulin sensitivity in overweight adults
Lower LPS-dependent chronic low-grade inflammation	Increase LPS-suppressing bacteria and decrease LPS-producing bacteria	Reductions in liver fat and systemic inflammation in adults with non-alcoholic fatty liver disease
Reduce metabolic, inflammatory, and oxidative stress biomarkers	Promote SCFA production	Improved quality of life and inflammatory markers in colorectal cancer patients

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
