# Peer review of "The Potential Effects of Probiotics and ω-3 Fatty Acids on Chronic Low-Grade Inflammation"

_nutrients, 2020, doi:10.3390/nu12082402_

Round 1

Reviewer 1 Report

Authors decided to examine whether 1: additive effects of n=3 pUFA supplementation exist with probiotics on immune regulation and inflammation in older or obese individuals and 2: gut-brain connumication

Authros argue that existance of inflammation and aging correlatively affect adverse health outcomes and that modulation of gut microbiome could be utilized to counter this imbalance and lastly ecistanc eof gut-brain axis mediated by circulating SFAs, neurotransmitters and other metabolites that are intestinally derived. Finally, to fish off the review, the authors discuss therapeutic intervention by probiotics and/or n-3 PUFA and potential synergism between the two. Although logical and concise, it is speculative on the scientific background to support additive benefits of the two, hence th title should be reworded to " The potential effects of" to reflect the nature of data available.

Author Response

Response to Reviewer 1 comments

Authros argue that existance of inflammation and aging correlatively affect adverse health outcomes and that modulation of gut microbiome could be utilized to counter this imbalance and lastly ecistanc eof gut-brain axis mediated by circulating SFAs, neurotransmitters and other metabolites that are intestinally derived. Finally, to fish off the review, the authors discuss therapeutic intervention by probiotics and/or n-3 PUFA and potential synergism between the two. Although logical and concise, it is speculative on the scientific background to support additive benefits of the two, hence th title should be reworded to " The potential effects of" to reflect the nature of data available.

Thank you for reviewing our manuscript and for the insightful comments.  While we believe that there is evidence to support the additive effects of the two supplements, we agree with the reviewer that this claim, especially in the title, may be too strong and speculative.  We appreciate the suggestion to change the title.  We have changed the title from "The effect of probiotics..." to "The potential effects of probiotics..." as the reviewer has suggested (Line 2 and 3).  

Reviewer 2 Report

The contents presented in the review manuscript are original and interesting regarding the effect of probiotics and ω-3 Fatty Acids on chronic low-grade inflammation. This article comprehensively summarized molecular mechanisms and therapeutic potentials of supplements, prebiotics, and probiotics on inflammatory and/or psychiatric diseases.

General Comments

This study is well-written for readers to understand the importance of gut bacteria to maintenance of homeostasis of the body and brain. The manuscript is, for the most part, clearly presented and concisely summarized with high degree of completion. There is few or no point that the authors need to modify.

Specific Comments

  • Major comment
  1. The reviewer feels that introduction section is lengthy for the content, especially in line 58-69 about inflammaging. If possible, the authors should make this part more concise.

Author Response

Response to Reviewer #2

General Comments

This study is well-written for readers to understand the importance of gut bacteria to maintenance of homeostasis of the body and brain. The manuscript is, for the most part, clearly presented and concisely summarized with high degree of completion. There is few or no point that the authors need to modify.

Specific Comments

  • Major comment
  1. The reviewer feels that introduction section is lengthy for the content, especially in line 58-69 about inflammaging. If possible, the authors should make this part more concise.

Thank you for reviewing our manuscript and for the comments.  We are pleased to find that there were few or no points that we needed to modify.  We agree that perhaps too much detail about inflammaging is presented in the introduction.  Therefore, we have condensed this description by removing some of the detail about the conditions that inflammaging can lead to.  These changes can be located in line 67 of the text.  

Reviewer 3 Report

In the review “The Effect of Probiotics and ω-3 Fatty Acids on Chronic Low-Grade Inflammation” the authors reviewed the effect of probiotics and ω-3 supplementation on chronic low-grade inflammation of the gut and its impact on the gut-brain axis communication. The authors present several controlled trials, where the relation of probiotics, ω-3, and probiotics+ω-3 with the inflammatory status is studied.  The review is well written and very easy to follow, however, some minor questions should be addressed:

  • Please explain in more detail how Lactobacilli and Bifidobacteria, among others, prevent gut hyperpermeability.
  • Please increase the quality of Figure 1.
  • Page 3, line 119 – please revise.
  • A Table resuming the beneficial effects of probiotics, ω-3, and probiotics+ω-3 will increase the overall quality of the review and will help the readers.

Author Response

Response to Reviewer #3

In the review “The Effect of Probiotics and ω-3 Fatty Acids on Chronic Low-Grade Inflammation” the authors reviewed the effect of probiotics and ω-3 supplementation on chronic low-grade inflammation of the gut and its impact on the gut-brain axis communication. The authors present several controlled trials, where the relation of probiotics, ω-3, and probiotics+ω-3 with the inflammatory status is studied.  The review is well written and very easy to follow, however, some minor questions should be addressed:

  • Please explain in more detail how Lactobacilli and Bifidobacteria, among others, prevent gut hyperpermeability.

More detail has been included about the mechanisms by which these strains affect gut permeability.  We agree that additional details strengthen this section.  

  • Please increase the quality of Figure 1.

We agree that the quality of Figure 1 could be improved.  Here we have attached a file containing Figure 1 with higher quality. 

  • Page 3, line 119 – please revise.

Thank you for pointing out the error with the font size in this line.  It has been changed from size 9 to size 10 to be consistent with the rest of the manuscript.  

  • A Table resuming the beneficial effects of probiotics, ω-3, and probiotics+ω-3 will increase the overall quality of the review and will help the readers.

A table has now been included at the end of the conclusion section (line 403) to summarize the beneficial effects of each supplement individually as well as when they are combined.  
